# Niclosamide targets the dynamic progression of macrophages for the resolution of endometriosis in a mouse model

Liang Zhao[1], Mingxin Shi[1], Sarayut Winuthayanon [1], James A. MacLean II [1] & Kanako Hayashi [1✉]

Due to the vital roles of macrophages in the pathogenesis of endometriosis, targeting macrophages could be a promising therapeutic direction. Here, we investigated the efficacy of niclosamide for the resolution of a perturbed microenvironment caused by dysregulated macrophages in a mouse model of endometriosis. Single-cell transcriptomic analysis revealed the heterogeneity of macrophages including three intermediate subtypes with sharing characteristics of traditional "small" or "large" peritoneal macrophages (SPMs and LPMs) in the peritoneal cavity. Endometriosis-like lesions (ELL) enhanced the differentiation of recruited macrophages, promoted the replenishment of resident LPMs, and increased the ablation of embryo-derived LPMs, which were stepwise suppressed by niclosamide. In addition, niclosamide restored intercellular communications between macrophages and B cells. Therefore, niclosamide rescued the perturbed microenvironment in endometriosis through its fine regulations on the dynamic progression of macrophages. Validation of similar macrophage pathogenesis in patients will further promote the clinical usage of niclosamide for endometriosis treatment.

[1] School of Molecular Biosciences, Center for Reproductive Biology, Washington State University, Pullman, Washington 99164, USA. ✉email: k.hayashi@wsu.edu

Endometriosis is a common chronic inflammatory disease that affects roughly 10% of reproductive-aged and adolescent women worldwide[1,2]. It is characterized by the presence and growth of tissues resembling endometrium, termed endometriotic lesions, outside of the uterus. Patients with endometriosis exhibit symptoms of chronic pelvic pain, infertility, and multiple other health issues leading to tremendous reductions in their quality of life[1]. Unfortunately, public and professional awareness of this disease remains poor. Current hormonal therapies, along with laparoscopic surgery, do not cure the disease and are often of limited efficacy with high recurrence rates, frequent side effects, and potential morbidity. Thus, a critical need exists to develop new and effective therapies for endometriosis targeting biologically important mechanisms that underlie the pathophysiology of this disease.

Disruption of the immune homoeostasis in the peritoneal cavity drives the disease development of endometriosis, and macrophages play a central role in this process[3–5]. Peritoneal macrophages infiltrate endometriotic lesions and promote their growth and vascularization by releasing proinflammatory cytokines and growth factors[5–8]. In addition, IGF1 and netrin-1, along with cytokines secreted by macrophages, also promote neurogenesis and innervation at lesion sites, which enhances the pain sensation in patients[9–12]. The proinflammatory cytokines released by macrophages and disrupted in endometriosis also affect multiple important activities of reproduction, such as hormonal balances and decidualization, leading to infertility[13,14]. Suppressing the release of proinflammatory cytokines and growth factors from macrophages inhibits lesion growth and endometriosis-associated pain in rodent models[5,6,9,15]. Therefore, targeting specific subtypes of peritoneal macrophages that are critical for maintaining immune homoeostasis in the pelvic cavity could be a new direction for drug development in endometriosis therapy.

To fully characterize the role of peritoneal macrophages in the pathophysiology of endometriosis, a better understanding of the heterogeneity of macrophage populations and their subtype-specific contributions to endometriosis is necessary. Two subsets of macrophages have previously been characterized in the peritoneal cavity and are referred to as "small" (SPMs) and "large" peritoneal macrophages (LPMs)[16]. MHC II$^{high}$ F4/80$^{low}$ SPMs are short-lived and are recruited from Ly6C$^+$ classical monocytes, while MHC II$^{low}$ F4/80$^{high}$ LPMs are resident and long-lived with an embryonic origin[17,18]. The population of embryo-derived resident LPMs uniquely express TIM4, and its number is mainly maintained through its self-renewal under physiological conditions[19]. With mild inflammation, some of the recruited SPMs gradually differentiate into F4/80$^{high}$ macrophages but still remain in an immature state due to the existence of enduring resident LPMs[18]. When the population of those resident LPMs is ablated with extended inflammation, these transitory F4/80$^{high}$ macrophages finally mature and replenish the pool of resident LPMs[18,20]. However, this newly recruited F4/80$^{high}$ macrophage population shows striking functional differences from those embryo-derived resident LPMs and thus increasing the risks for the incidence and severity of diseases in the future[18]. In endometriosis, dynamic and progressive alterations of peritoneal macrophages were also found associated with lesion development[21,22]. However, the transcriptomic characteristics of peritoneal macrophages especially those transitory subtypes and the molecular signalling networks that coordinate the dynamic progression of macrophages in endometriosis are unknown.

Niclosamide is an FDA-approved anthelmintic drug with multiple clinical trials ongoing to repurpose it for the treatment of other diseases including cancer and metabolic diseases[23]. We previously reported that niclosamide reduced lesion growth, alleviated aberrant inflammation in peritoneal fluids, and decreased the vascularization and innervation in lesions using two distinct mouse models of endometriosis[5,24].

In this study, we further focused on the heterogeneity of peritoneal macrophages and molecular mechanisms regulating their dynamic progression after lesion induction using a mouse model of endometriosis. Moreover, we found that niclosamide finely reversed those transcriptomic changes of macrophages caused by lesion induction through its stepwise regulations on the differentiation of recruited macrophages, the maturation of transitory LPMs, and the preservation of embryo-derived resident LPMs. Niclosamide also rescued the communications between resident LPMs and B cells which had been disrupted by lesion induction. Therefore, we propose that macrophages could be the direct target of niclosamide. Though more studies are in need to validate similar functions of niclosamide for the pathogenesis of macrophages in human patients, niclosamide showed promising effects for the treatment of endometriosis. Finally, to share our scRNA data with other researchers, we have created a cloud-based web tool (Webpage: https://kanakohayashilab.org/hayashi/en/mouse/peritoneal.immune.cells/) for the gene of interest searches that can be easily conducted without the requirement of complicated computer programming skills.

## Results

**Single-cell transcriptomic sequencing of peritoneal immune cells.** In this study, ELL were induced by inoculating menses-like tissues from donor mice into the peritoneal cavity of the recipient, as described in the Method section. Three weeks later, one group of mice was administrated niclosamide (ELL_N) while the others (sham and ELL) were given a control vehicle (Fig. 1a). After another three weeks of treatment, cells in the peritoneal cavity, mostly immune cells, were collected and processed for single-cell transcriptomic analysis. A total of 13,679 cells with a median of 3160 genes per cell were retained for downstream analysis after quality control and removal of low-quality cells.

Integrated cells from all three groups of samples were classified into 19 clusters based on the unsupervised clustering workflow of the Seurat package with cell identities determined by canonical marker gene distributions (Fig. 1b and Supplementary Fig. 1a). Cells from each group showed a consistent distribution within each cluster in UMAP (Supplementary Fig. 1b), suggesting an unbiased capture of cell populations between groups of different treatments. Macrophages (43%) and B cells (46%) are the most abundant populations identified in peritoneal fluids along with much fewer T cells and other immune cells (Fig. 1b, c).

**Heterogeneity of peritoneal macrophage populations.** Seven sub-clusters of macrophage-related populations (Pre-SPMs, SPMs, IM1, IM2, IM3, *Timd4*$^+$ LPMs, and PMs) were identified in these samples (Fig. 1b, c). Clusters of premature (Pre-SPMs; Cd226$^{low}$) and mature "small" peritoneal macrophages (SPMs; Cd226$^{high}$) express high levels of characteristic monocyte-derived cell markers including *H2-Aa* (which encodes a part of the MHCII), *Irf4* and *Ccr2*[17,18]. These two clusters are further distinguished by the expression of *Cd209a* and *Itgax* (CD11c) in Pre-SPMs and higher expression of *Cd226*, *Aif* and *Retnlα* (also known as *Relmα* or *Fizz1*) in SPMs (Fig. 1d and Supplementary Fig. 1a)[17]. The identity of *Timd4*$^+$ "large" peritoneal macrophages (*Timd4*$^+$ LPMs) was determined by the expression of the unique marker, *Timd4* (which encodes TIM4, Fig. 1d and S1a), for the embryonic-derived resident macrophages, which also express high levels of *Adgre1* (encodes F4/80). In addition, a group of proliferating macrophages (PMs) was identified by their exclusive

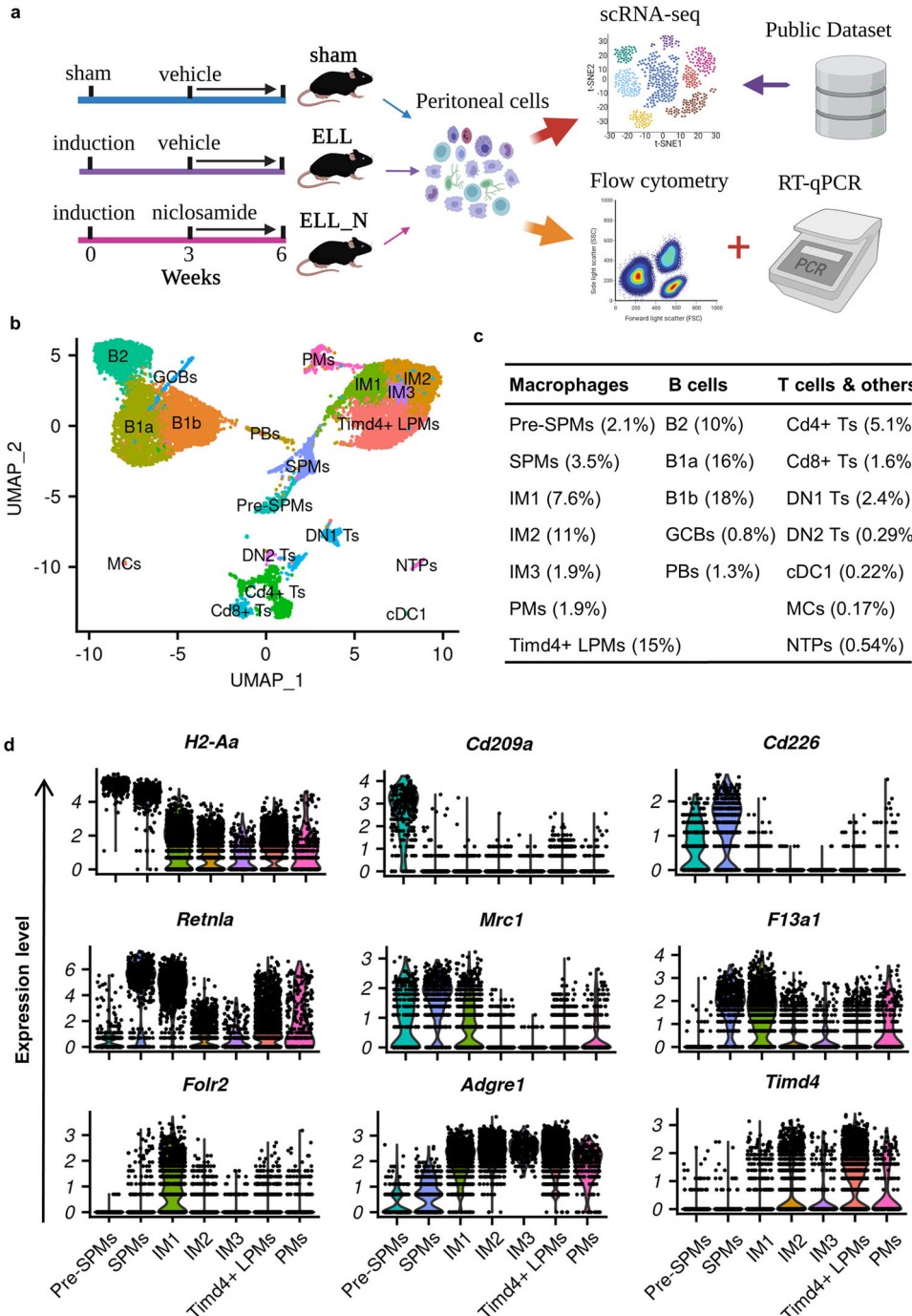

**Fig. 1 Single-cell transcriptomic profiling of peritoneal immune cells. a** Schematic of the experimental pipeline. Created with BioRender.com **b** UMAP visualization of peritoneal immune cells. **c** Ratio of each population in cell number. **d** VlnPlot of characteristic gene expressions for macrophage subpopulations. Sham sham control, ELL endometriosis-like lesions, ELL_N niclosamide administration to ELL-induced mouse, Pre-SPMs premature "small" peritoneal macrophages, SPMs "small" peritoneal macrophages, IM1-3 intermediate macrophages subtype 1-3, Timd4+ LPMs Timd4+ "large" peritoneal macrophages, PMs proliferating macrophages, B1a B1a cells, B1b B1b cells, B2 B2 cells, GCBs germinal-centre B cells, PBs plasma blast B cells, Cd4+ Ts Cd4+ T cells, Cd8+ Ts Cd8+ T cells, DN1 double negative T cells 1, DN2 double negative T cells 2, cDC1 conventional dendritic cells 1, MCs Mast cells, NTPs neutrophils.

expression of the proliferation marker, *Mki67* (Supplementary Fig. 1a).

In addition to these well-known types of peritoneal macrophages, three novel subtypes were also identified, which are considered differentiating intermediates and named intermediate macrophages (IM1-3). IM1 cells showed unique expression of *Folr2* (encodes folate receptor β (FRβ) subunit) and high levels of

*Trem2* (Fig. 1d and Supplementary Fig. 1a). Though cells of this cluster showed intermediate expression of the LPM marker *Adgre1*, equivalent levels of *Retnla*, *Mrc1* (CD206), *F13a1*, and *Aif1* as cells of the SPMs were also found (Fig. 1d and Supplementary Fig. 1a). These gene expression characteristics of IM1 suggest their close developmental relations to SPMs. The other two intermediate groups (IM2 and IM3) showed high

expression of *Adgre1*, but they are low in the expression of *Timd4*, indicating that they are newly recruited immature LPMs.

Next, the top 100 differentially expressed genes in each cluster were used to enrich their unique characteristics by gene ontology analysis (GO) of biological processes (Supplementary Fig. 2 and Supplementary Data 3). Up-regulated biological processes related to TNF production were found in all three intermediate subtypes (IM1, IM2, and IM3) based on enriched terms of "regulation of tumour necrosis factor production" in IM1, "positive regulation of tumour necrosis factor production" and "negative regulation of transforming growth factor beta production" in IM2, and "tumour necrosis factor production" in IM3. Different from these intermediate subtypes, one term of "regulation of transforming growth factor beta production" was enriched in *Timd4*+ LPMs, suggesting differential functions between newly recruited and embryo-derived macrophages. In addition, the three intermediate groups all show molecular signatures characteristic of "phagocytosis" or "cell killing", which were not found in the population of *Timd4*+ LPMs. Different from Pre-SPMs and SPMs, enriched biological processes to support the "regulation of angiogenesis" were found in the cells of IM1, IM2, IM3, and *Timd4*+ LPMs. This high-resolution analysis of macrophage transcriptomes identified in our study provides us with an unprecedented opportunity to study stage-specific effects caused by ELL and the treatment of niclosamide.

**Peritoneal macrophages in a normal physiological state**. A publicly-available single-cell RNA-seq dataset from CD11b+ peritoneal macrophages in wild-type female mice in a normal physiological state [GSM4151331[25]] was re-analysed in this study (Supplementary Fig. 3a, b). Similar subpopulations of "Pre-SPM", "SPM", "IM" and four clusters of *Timd4*+ embryo-derived LPMs (LPM1-4) were identified. However, different from our samples collected from Sham, ELL, and ELL_N groups, no *Timd4*- *Adgre1*high (corresponding to IM2 and IM3 of our study) subtypes were distinguished in these macrophages at a presumably baseline physiological state.

**Niclosamide reverses lesion-induced transcriptomic changes in macrophages**. Transcriptomic changes in the macrophages induced by ELL and following niclosamide treatment (ELL_N) were compared by gene set enrichment analysis (GSEA). Compared to the sham group, a total of 135 biological processes were up-regulated by ELL induction (Fig. 2a and Supplementary Data 4). Of these processes, 78 were reversed by niclosamide (Fig. 2a and Supplementary Data 4). More specific, ELL induced activation, proliferation, and differentiation of peritoneal macrophages as GO terms of "Establishment or maintenance of cell polarity", "Transmembrane receptor protein tyrosine kinase signalling pathway", "Lymphocyte proliferation", "Positive regulation of cell migration" and "Regulation of lymphocyte differentiation" were all positively enriched compared to the sham group (Fig. 2d). ELL also up-regulated biological processes of "Vesical organization", "Ceramide transport", "Regulation of neuron differentiation" and "Angiogenesis", which are associated with vascularization, neurogenesis, and pain sensation in lesions. All of the biological processes above were suppressed by niclosamide compared to the ELL group (Fig. 2d). Conversely, 10 out of 12 ELL-inhibited pathways were enhanced by niclosamide (Fig. 2b and Supplementary Data 4). For example, niclosamide promoted biological processes of "Cytoplasmic translation" and "Oxidative phosphorylation" which were reduced by ELL induction (Fig. 2e and Supplementary Data 4).

Niclosamide tunes disrupted macrophages back to a relevant homoeostatic level with the sham group after 3 weeks of

treatment (ELL_N/sham), with only 25 differential regulated pathways found (Supplementary Data 4 sheet "ELL_N to sham"). Compared to the sham group, niclosamide further decreased the inflammatory responses and oxidative stress in macrophages but promoted their apoptosis as indicated by enriched GO terms of "Macrophage derived foam cell differentiation", "I-kappaB kinase/NF-kappaB signalling", "Cellular response to oxidative stress" and "Negative regulation of apoptotic signalling pathways" (Fig. 2c). These results indicate that ELL induced aberrant activation of macrophages and enhanced their signalling communications for lesion growth and pain sensation, which were finely reversed by niclosamide at the transcriptomic level.

**Niclosamide suppressed the expression of genes that were enhanced by ELL**. To further understand the transcriptomic changes in macrophages caused by ELL and niclosamide, we examined their corresponding alterations at the gene level (Fig. 3a). A total of 114 genes were up-regulated by ELL compared to the sham group (ELL/sham) with 91 of them being decreased by niclosamide (ELL_N/ELL). These top up-regulated genes by the presence of ELL include *Retnla*, *Mrc1*, *F13a1*, *Kctd12*, *Plxnd1*, *Ccl9*, *Ccl6*, and Socs6 ($p < 5.1e-8$), which were all inhibited by niclosamide (Fig. 3c). These differentially expressed genes were also confirmed by qPCR analyses in independent samples (Fig. 3d). Consistently, ELL enhanced the expression of *Retnla* ($p = 0.008$), *Mrc1* ($p = 0.030$), *Ccl6* ($p = 0.035$), *Kctd12* ($p = 0.055$), and *Scocs6* ($p = 0.015$) while niclosamide suppressed the expression of *Retnla* ($p = 0.004$), *F13a1* ($p = 0.047$), *Mrc1* ($p = 0.024$), *Ccl6* ($p = 0.056$), *Kctd12* ($p = 0.049$), *Scocs6* ($p = 0.022$), and *Plxnd1* ($p = 0.030$). Interestingly, most of these up-regulated genes by ELL were uniquely expressed in the populations of Pre-SPMs, SPMs, and IM1 (Fig. 3e) but not the others. Consistent with their distributions in our samples, a similar pattern was also found in the re-analysed dataset of macrophages at the normal physiological state (Pre-SPM, SPM, and IM1; Supplementary Fig. 3c). Therefore, ELL enhanced gene expressions related to the lineage of monocyte-derived cells, and niclosamide reversed those changes.

**Niclosamide attenuates the maturation of "small" peritoneal macrophages**. Cells from Pre-SPMs, SPMs, and IM1 showed gene expression characteristics of monocyte-derived macrophages including expression of *H2-Aa*, *Retnlα*, *Mrc1*, *Irf4*, *Cd226*, *Ccr2* and *Aif* (Fig. 1d and Supplementary Fig. 1a). Previous studies also showed that increased levels of CD226 were required for the gradual maturation of SPMs from monocyte-derived cells in an IRF4-dependent manner[17]. Therefore, the differentiation of cells from Pre-SPMs to SPMs corresponds to the maturation process of SPMs from monocyte-derived cells. Moreover, the population of IM1 was recently reported to be immediately derived from CCR2+ monocyte-derived macrophages[25], which correspond to cells of Pre-SPMs and SPMs in this study. Using flow cytometry, we also found that about 25% of FRβ+ macrophages still retained high expression of Ly6C on day 7 after ELL induction with a significant decrease on day 42 ($p = 0.025$; Supplementary Fig. 4a), which further confirmed the monocyte-origin of this cell population. The ratio of the MHCII+ subset of FRβ+ macrophages was also decreased from day 7 to 42 ($p = 0.044$), suggesting that the recruited FRβ+ macrophages gradually lost their monocyte-related characteristics and became a more differentiated subtype over time (Supplementary Fig. 4a). Therefore, the development of cells from Pre-SPMs to SPMs and IM1 is considered a continuous process of the maturation of SPMs from monocyte-derived cell and their further differentiation into the intermediate subtype

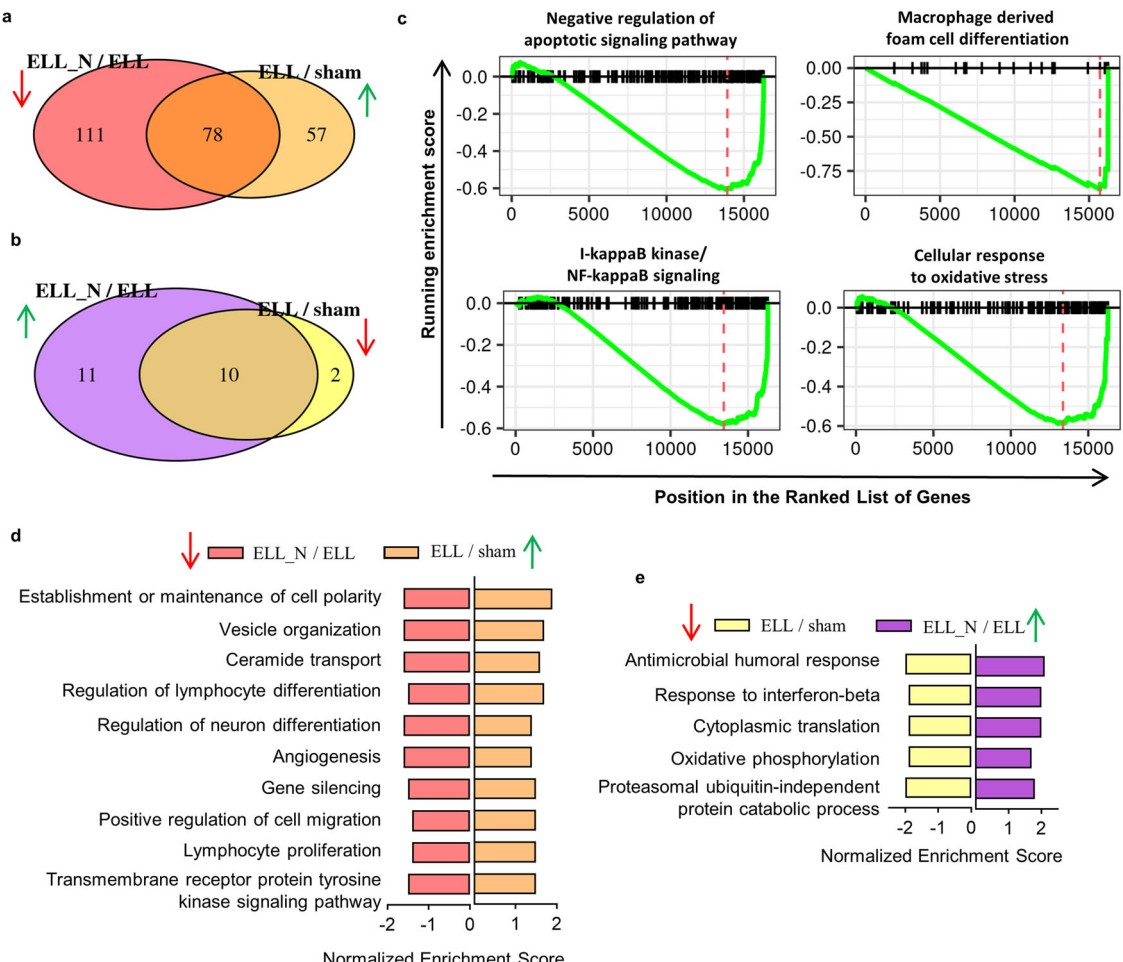

**Fig. 2 Niclosamide finely reverses transcriptomic changes caused by endometriosis-like lesions (ELL) to peritoneal macrophages. a** Venn diagram shows the overlaps of enriched GSEA terms of biological processes between those enhanced in by ELL (ELL/sham) and those reduced by niclosamide (ELL_N/ELL). **b** Similar to (**a**) but shows the overlaps between enriched terms of those down-regulated in ELL (ELL/sham) and up-regulated by niclosamide (ELL_N/ELL). **c** Enriched GSEA terms of biological processes that were reduced within the ELL_N group compared to the sham group. **d** Representative GSEA terms that were enhanced by ELL (ELL/sham) and were decreased by niclosamide (ELL_N/ELL). **e** Representative GSEA terms that were inhibited by ELL (ELL/sham) and were reversed by niclosamide (ELL_N/ELL).

with sharing characteristics of both SPMs and LPMs (*Folr2*$^{high}$ *Adgre*$^{intermediate}$ IM1).

We next applied pseudo-temporal trajectory analysis to these three subpopulations to elucidate the dynamic changes and functions of ELL-enhanced genes along this early differentiation process of recruited macrophages. Cells from Pre-SPMs, SPMs, and IM1 were computationally selected, and a continuous trajectory of the differentiation process was constructed using Monocle3 (Fig. 4a). The expression of genes enhanced by ELL including *Retnla*, *Mrc1*, *F13a1*, *Kctd12*, *Plxnd1*, *Ccl9*, *Ccl6*, and *Socs6* (Fig. 3c, d) were plotted along this developmental timeline (Fig. 4b). Interestingly, most of these genes showed a consistent upregulation pattern during the early maturation process of SPMs from Pre-SPMs (Fig. 4b). This dynamic expression pattern was also confirmed by the re-analysed datasets of normal macrophages as described above (Supplementary Fig. 5a, b).

Among those genes, the expression of *Retnla* increases by about 100-folds during this process of differentiation (Fig. 4b). The function of *Retnla* for this dynamic process was further studied by in silico knockout *Retnla* in these three subpopulations. The top dysregulated genes by virtual knockout of *Retnla* include *Gas6*, *H2-Oa*, *Cd209a*, *Ccr2*, *Il1b*, and *Folr2*

(Supplementary Data 5). Those perturbed genes affected multiple important pathways of immune responses such as biological processes of "leukocyte migration", "leukocyte chemotaxis", "phagocytosis", "regulation of cytokine production involved in immune response", "antigen processing and presentation" and "regulation of interleukin-2 production" (Fig. 4c and Supplementary Data 5). In addition, *Retnla* knockout may also affect the communications between macrophages and T cells, shown by GO terms of "T cell activation" and "T cell differentiation".

As ELL enhanced the expression of genes that are necessary for the differentiation of SPMs from Pre-SPMs, an increased population number of SPMs was expected in the ELL group. By the analysis of flow cytometry, we confirmed a more than 3 times increase in the number of Ly6C$^+$ CD11b$^+$ monocyte-derived macrophages in the ELL group ($p = 0.001$), which was reduced to a similar level as the sham group by niclosamide (Fig. 4d). As a consequence, the populations of FRβ$^+$ (encoded by *Folr2*) IM1 ($p = 0.005$) and CD206$^+$ (encoded by *Mrc1*; $p = 0.025$) macrophages were also increased by ELL and decreased by niclosamide ($p = 0.003$ and $p = 0.0004$, respectively; Fig. 4e). Therefore, niclosamide attenuated the recruitment of monocyte-derived macrophages by decreasing its maturation from Pre-SPMs, which was promoted by ELL.

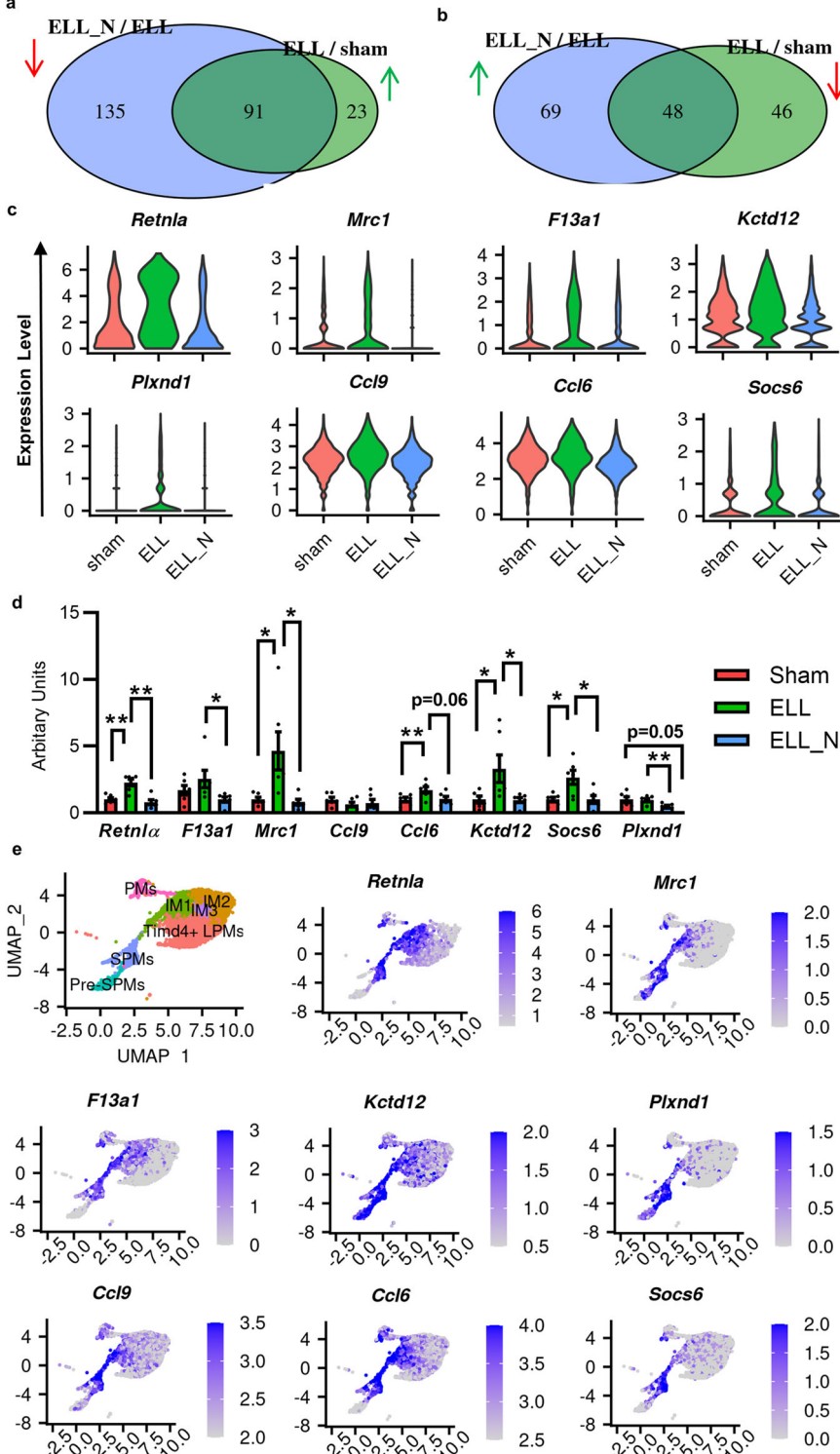

**Fig. 3 Genes regulated by ELL and niclosamide and their distributions within the perineal macrophage subpopulations (suppressed by niclosamide).**
**a** Venn diagram shows that overlaps of genes that were enhanced by ELL (ELL/sham) but decreased by niclosamide (ELL_N/ELL). **b** Overlaps of genes that were inhibited by ELL (ELL/sham) but enhanced by niclosamide (ELL_N/ELL). **c** Vlnplot shows representative genes that were enhanced by ELL but were reduced by niclosamide treatment (ELL_N). All genes shown here were significantly ($p < 0.05$) differentially expressed as determined by the "wilcox" test within the Seurat package. **d** Verification of differential gene expression by RT-qPCR. *$p < 0.05$, **$p < 0.01$, mean ± SEM, $n = 6$ per group. **e** UMAP of computationally selected macrophage populations and the distribution of genes within them.

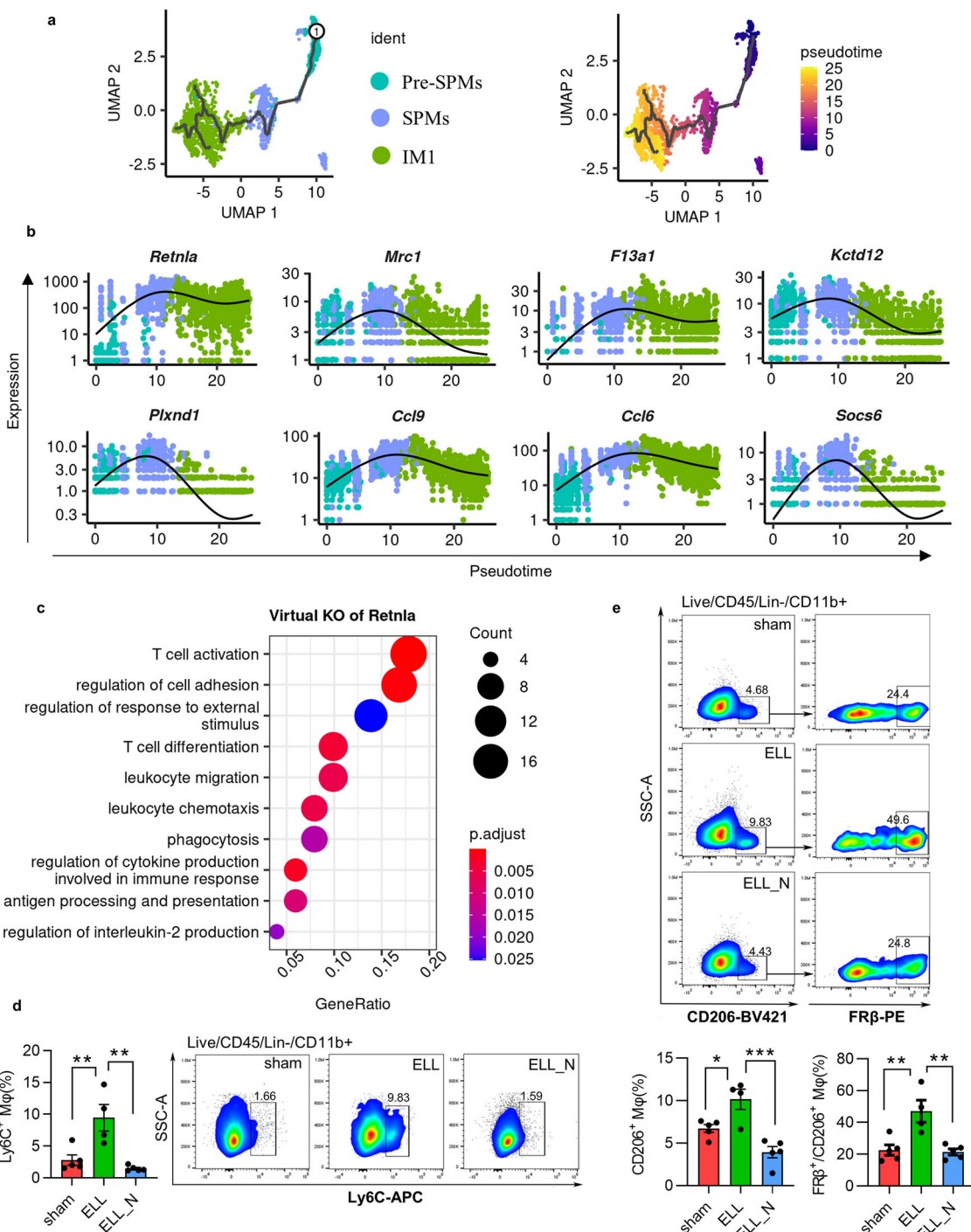

**Fig. 4 Reconstruction of a pseudo-temporal trajectory for the early maturation and differentiation of recruited macrophages. a** UMAP shows selected cells of recruited macrophages (left) and a trajectory path built by Monocle 3 (right). The selected root cell was labelled as ①. **b** Dynamic changes of genes along this trajectory path of differentiation. **c** GO terms of biological processes that were enriched by perturbed genes affected by virtual KO of *Rentla*. **d** Flow cytometer isolation and quantification of Ly6C+ recruited monocytes. **e** Flow cytometer results and quantification of CD206+ and FRβ+ macrophages. *$p < 0.05$, **$p < 0.01$, ***$p < 0.001$, mean ± SEM, $n = 5$ per group.

**Niclosamide increased the expression of genes that were decreased by ELL.** In addition to niclosamide's suppressions on the genes that were enhanced by ELL, niclosamide also up-regulated over 50% of genes that were reduced by ELL induction (Fig. 3b). The top representative genes include *Cfb, Hp, Ifitm2, Ifitm3, Gbp2b, C1qb, Prdx5*, and *Gngt2* ($p < 6.8e-32$; Fig. 5a). The results of qPCR of immune cells in the peritoneal fluid with different treatments (Fig. 5b) showed consistent changes in the

expression of *Cfb* (ELL/sham, $p1 = 0.033$; ELL_N/sham, $p2 = 0.046$), *Ifitm2* ($p1 = 0.002$; $p2 = 0.03$), *Ifitm3* ($p1 = 0.020$; $p2 = 0.029$), *Gbp2b* ($p1 = 0.01$; $p2 = 0.04$), *C1qb* ($p1 = 0.005$; $p2 = 0.009$), *Prdx5* ($p1 = 0.002$; $p2 < 0.001$), and *Gngt2* ($p1 = 0.007$; $p2 < 0.002$). Interestingly, most of these genes were highly expressed in the intermediate subtypes of macrophages and *Timd4+* LPMs, but their expression in Pre-SPMs and SPMs was very low (Fig. 5c).

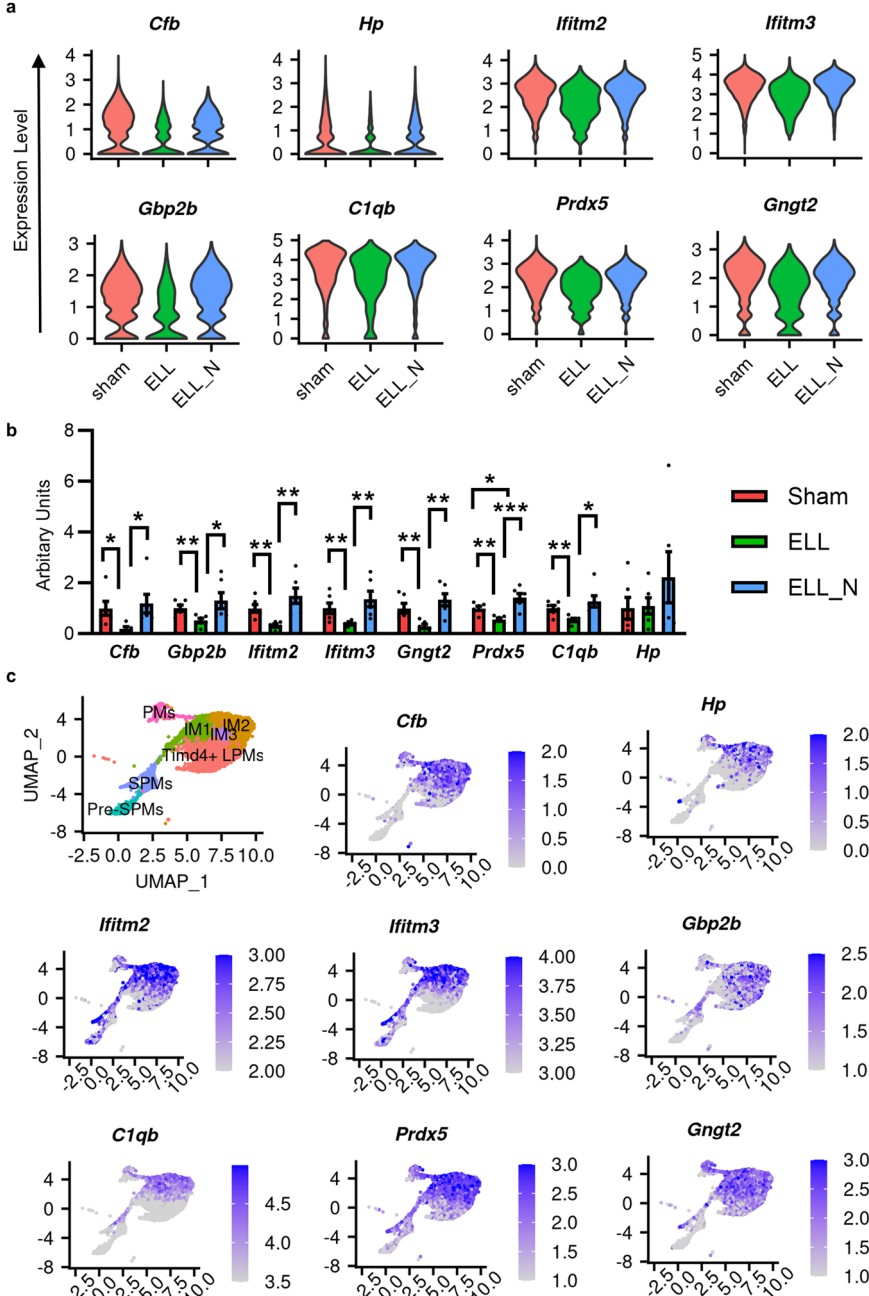

**Fig. 5 Genes regulated by ELL and niclosamide and their distributions within the perineal macrophage subpopulations (enhanced by niclosamide).**
**a** VlnPlot shows representative genes that were reduced by ELL (ELL/sham) but were enhanced by niclosamide (ELL_N/ELL). All genes shown here were significantly ($p < 0.05$) differentially expressed as determined by the "wilcox" test within the Seurat package. **b** Verification of differential gene expression by RT-qPCR. *$p < 0.05$, **$p < 0.01$, ***$p < 0.001$, mean ± SEM, $n = 6$ per group. **c** UMAP of computationally selected macrophage populations and the distribution of genes within them.

**Niclosamide decreased the maturation of recruited macrophages.** TIM4- F4/80[high] macrophages were reported to be monocyte-derived[18]. In support of this, $Ccr2^{-/-}$ mice ablate this population in the peritoneal cavity[18]. At a steady state, less than 5% of the TIM4- macrophage population replenish the TIM4+ embryo-derived macrophages of female mice[18]. However, mild or severe inflammation leads to dramatic reduction or ablation of TIM4+ LPMs, which promotes their replenishment by recruited TIM4- (Timd4) monocyte-derived macrophages[18,26]. Different from "IM1", cells of "IM2" and "IM3" do not share characteristics of "SPMs", with the less or minimum expression of $Retnla$, $Mrc1$, and $Folr2$, and instead acquire more characteristics of "LPMs"

including enhanced expression of $Adgre1$. Our new data also showed a sharp decrease of the TIM4+ F4/80[high] macrophage population on day 3 after lesion induction ($p < 0.0001$) known as a macrophage disappearance reaction, which then gradually increased from day 3 to day 42 ($p < 0.0001$; Supplementary Fig. 4b). However, these newly recruited "$Timd4$+ LPMs" have been reported to possess different functions from embryonic-derived "$Timd4$+ LPMs", which are associated with inflammation-driven altered differentiation[26]. Therefore, the three subtypes of $Adgre1^{high}$ macrophages, IM2, IM3, and $Timd4$+ LPMs (Fig. 6a) were involved in this biological process of maturation of $Timd4$- LPMs and the replenishment of $Timd4$+

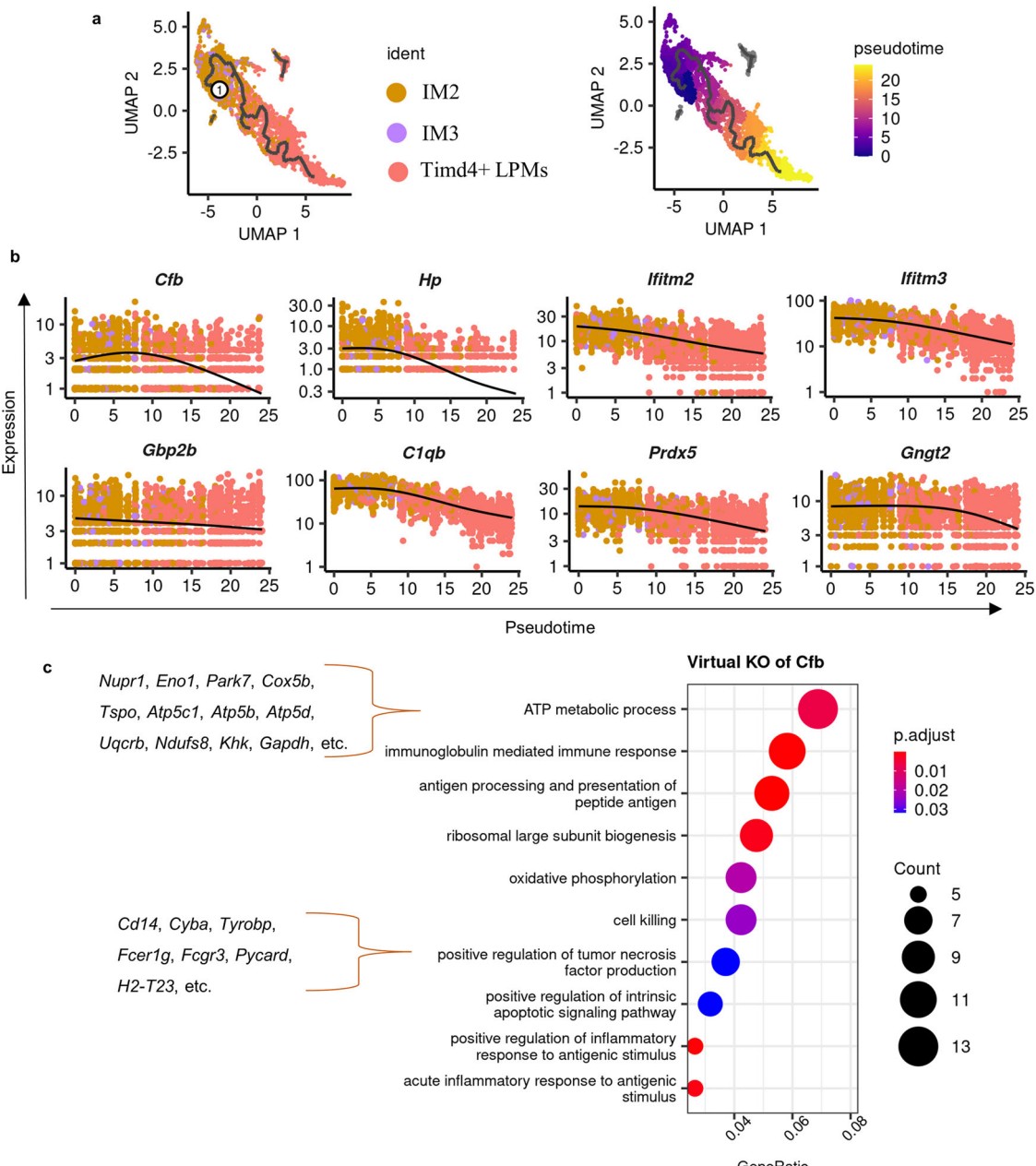

**Fig. 6 Reconstruction of a pseudo-temporal trajectory for maturation of intermediate "large" macrophages. a** UMAP shows selected cells of "large" macrophages (left) and a trajectory path built by Monocle 3 (right). The selected root cell was labelled as ①. **b** Dynamic changes of genes along this trajectory path of maturation and replenishment. **c** GO terms of biological processes that were enriched by perturbed genes affected by virtual KO of *Cfb*.

embryonic-derived macrophages. A pseudo developmental trajectory was thus reconstructed between them (Fig. 6a). The down-regulated genes caused by ELL including *Cfb*, *Hp*, *Ifitm2*, *Ifitm3*, *Gbp2b*, *C1qb*, *Prdx5*, and *Gngt2* (Fig. 5a) were plotted along this trajectory path (Fig. 6b). Interestingly, the expression of these genes showed a consistent decreasing pattern during this biological process, suggesting their important roles in maturation inducing.

As *Cfb* is one of the most responsive genes regulated by ELL and niclosamide in this biological process (Fig. 5a), we further explored its functions by in silico knockout of *Cfb* in cells of IM2, IM3, and *Timd4*⁺ LPMs (Supplementary Data 5). This analysis showed that virtual knockout of *Cfb* not only disrupted the inflammatory responses of macrophages but also changed the metabolism, protein synthesis, and apoptosis of macrophages as

terms of "ATP metabolic process", "Oxidative phosphorylation", "ribosomal large subunit biogenesis", and "positive regulation of intrinsic apoptotic signalling pathway" were enriched based on genes disrupted by *Cfb* knockout (Fig. 6c and Supplementary Data 5). Moreover, *Cfb* seems to be important for TNF production in these IM2 and IM3 based on enriched terms of "positive regulation on tumour necrosis factor production" (Supplementary Fig. 2).

As no *Timd4*⁻ *Forl2*⁻ intermediate macrophage phenotypes (IM2 and IM3 in our dataset) were identified in the publicly available dataset of normal macrophages, no obvious changes of genes mentioned above were found along the trajectory built between those LPMs (LPM1-4, Supplementary Fig. 6a, b). This difference also supports the previous finding that the maturation of macrophages is only active upon external stimuli[26]. Therefore,

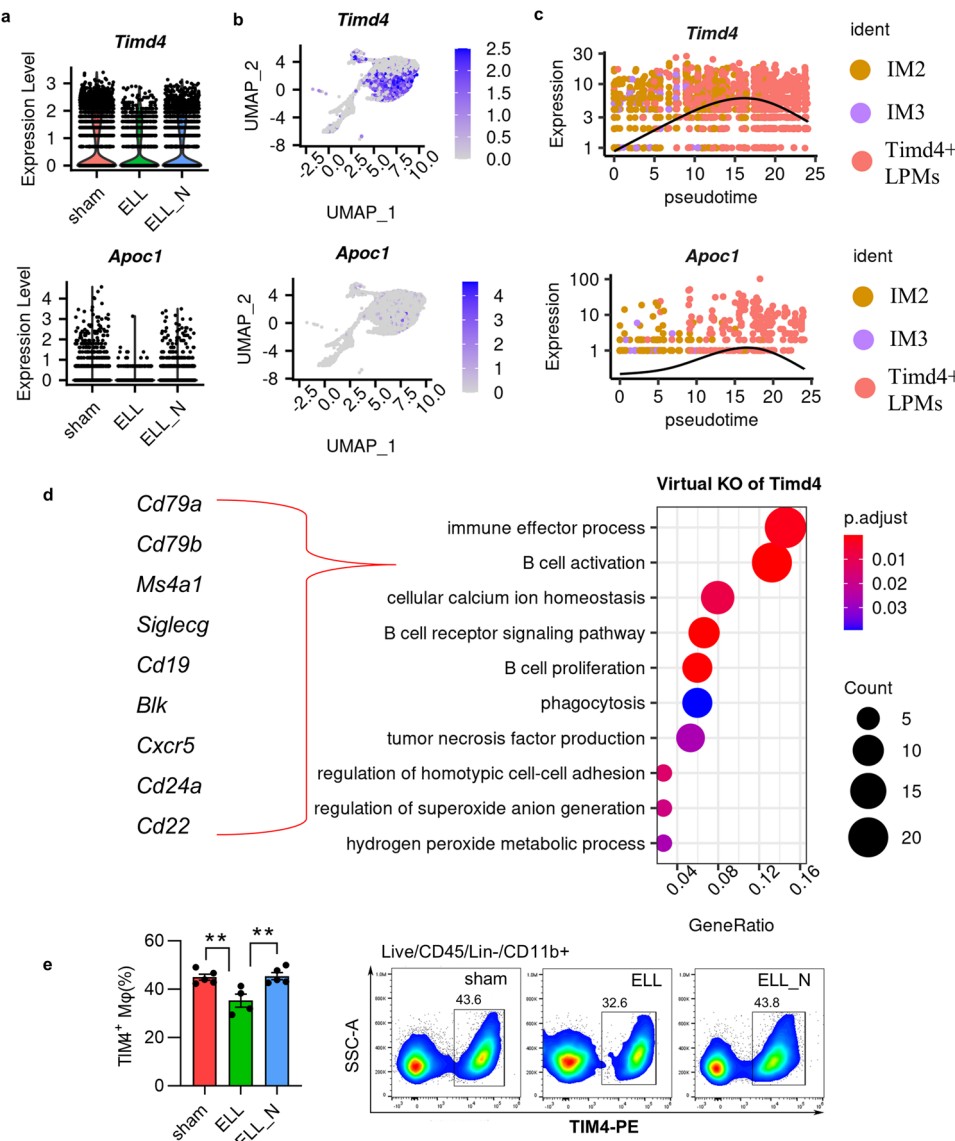

**Fig. 7 Niclosamide preserves the population of embryo-derived resident "large" peritoneal macrophages. a** VlnPlot shows gene expression of *Timd4* and *Apoc1*. All genes shown here were significantly (*p* < 0.05) differentially expressed as determined by the "wilcox" test within the Seurat package. **b** The featured plot shows the distribution of *Timd4* and *Apoc1*. **c** Dynamic changes of genes along the trajectory path of maturation and replenishment. **d** GO terms of biological processes that were enriched by perturbed genes affected by virtual KO of *Timd4*. **e** Flow cytometer isolation and quantification of TIM4+ resident macrophages. **p < 0.01, mean ± SEM, *n* = 5 per group.

ELL promotes the maturation of *Timd4*⁻ macrophages to replenish the resident embryonic-derived macrophage pool by downregulating genes such as *Cfb*, and this process is inhibited by niclosamide.

**Niclosamide reduces the ablation of embryo-derived resident macrophages**. The existence of TIM4+ embryo-derived LPMs prohibits the replenishment of the resident macrophage pool by recruited ones, while increased inflammation leads to ablation of embryo-derived LPMs and promotes the process of replenishment[26]. In support of this, ELL decreased the expression of *Timd4* and *Apoc1*, markers for embryo-derived LPMs, in the peritoneal macrophages (Fig. 7a, b). Expression of *Timd4* and *Apoc1* was also found to increase along with the initial maturation process of intermediate macrophages (Fig. 7c), suggesting recruited macrophages also gradually acquire their residency. Furthermore, knockout of *Timd4* in cells of IM2, IM3 and *Timd4*+ LPMs was shown to induce their B cell characteristics as

terms of "B cell activation", "B cell receptor signalling pathway" and "B cell proliferation" were enriched based on disrupted genes of *Timd4* knockout (Fig. 7d and Supplementary Data 5). Knockout *Timd4* also induced changes in TNF production, phagocytosis, and oxidative stress in those macrophages.

As a consequence of reduced *Timd4* expression in macrophages by ELL, the population number of TIM4+ LPMs was also reduced (*p* = 0.008; Fig. 7e). Niclosamide rescued the expression of *Timd4* (*p* = 2.5e-8) and also the number of TIM4+ LPMs (*p* = 0.006; Fig. 7e). Thus, ELL enhanced the replenishment of the embryo-derived macrophage pool by increasing the ablation of *Timd4*+ LPMs, which further promotes the maturation of recruited *Timd4*⁻ macrophages. Niclosamide preserves these embryo-derived LPMs by enhancing *Timd4* expression and recuses the homoeostasis of macrophages disrupted by ELL.

**Niclosamide rescues the communications between macrophages and B cells**. CXCL13-producing embryo-derived LPMs

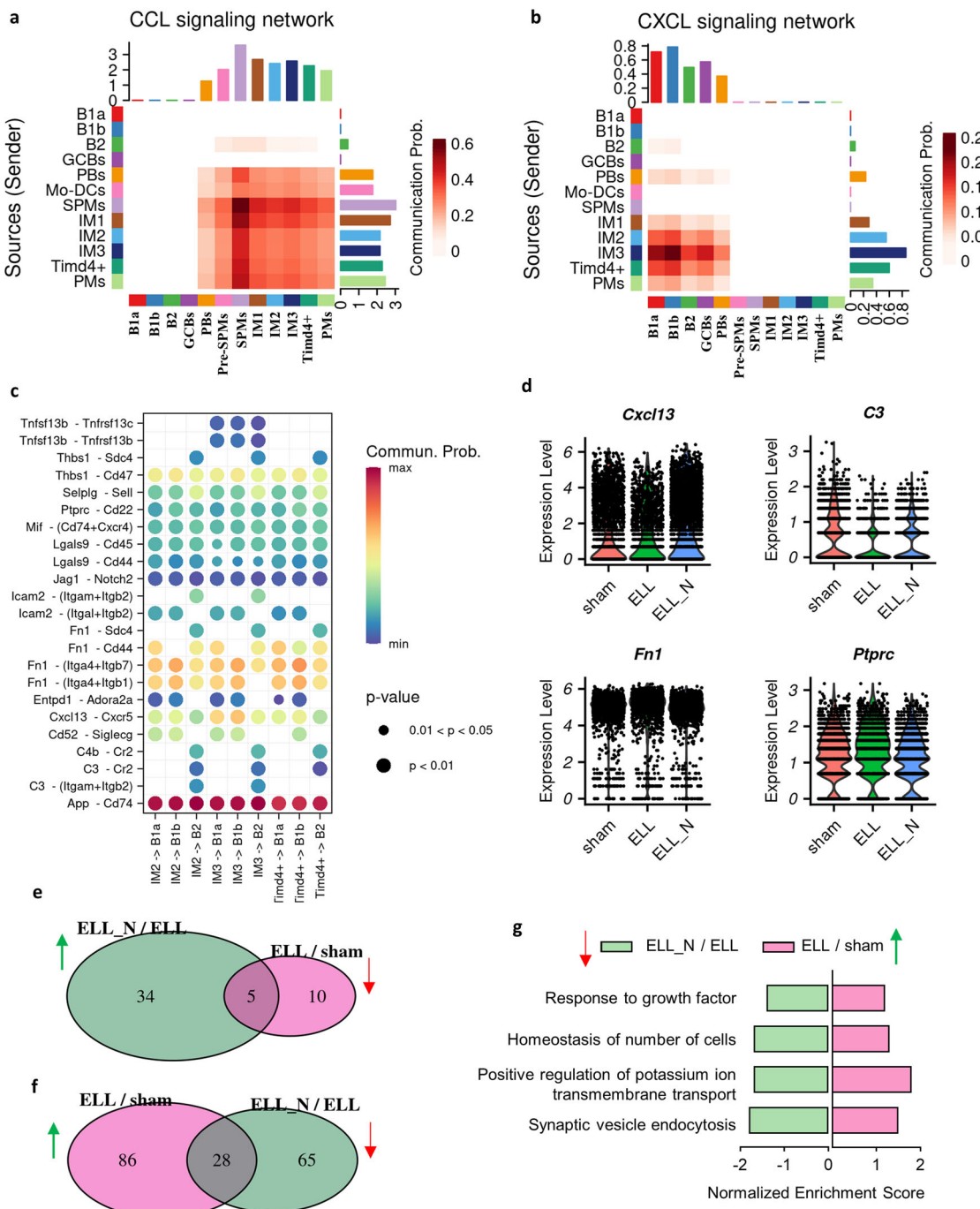

**Fig. 8 Niclosamide rescued the disrupted intercellular communications from macrophages to B cells. a** Heatmap showing the interactions between macrophages and B cells in terms of CCL signalling networks. **b** Heatmap showing the interactions between macrophages and B cells in terms of CXCL signalling networks. **c** Bubble plot showing all significant expressed ligand-receptor pairs identified between three subtypes of "large" macrophages (IM2, IM3, Timd4+ LPMs) and three subtypes of B cells (B1a, B1b, B2). **d** VlnPlot showing the differential expressed genes in macrophages. All genes shown here were significantly ($p < 0.05$) differentially expressed as determined by the "wilcox" test within the Seurat package. **e** Overlaps of GO biological processes in B cells that were suppressed by ELL (ELL/sham) but enhanced by niclosamide (ELL_N/ELL). **f** Overlaps of GO biological processes in B cells that were promoted by ELL (ELL/sham) but suppressed by niclosamide (ELL_N/ELL). **g** Overlaps of GO biological processes that were promoted by ELL by suppressed by niclosamide.

play an important role in the maintenance and recruitment of peritoneal B1 cells upon inflammation, while newly recruited macrophages were reported to be deficient in CXCL13[25–28]. To understand the communications between macrophages and B cells among their subpopulations, two important signalling networks for immune cell recruitment, CXCL and CCL, were analysed based on the ligand and receptor interactions between cells

using the CellChat package. Most of the CCL ligands were found to be released by macrophages and received by themselves, whereas there were very few communications between macrophages and B cells (Fig. 8a). The SPMs were the most affected cells by the CCL signalling among the other types of macrophages, which is consistent with their increased recruitment by ELL induction.

On the other hand, the expression of CXCL ligands was found to be mostly expressed in macrophages of IM2, IM3, and *Timd4*+ LPMs, and signals received by B cells (Fig. 8b). Furthermore, we explored the interactions between these CXCL-producing "large" types of macrophages and the three largest populations of B cells (B1a, B1b, and B2). Twenty-three significantly expressed ligand-receptor pairs were identified including *App-Cd74*, *Cxcl13-Cxcr5*, *C3-Cr2*, *Fn1-Sdc4*, *Ptprc-Cd22* (Fig. 8c), which may play important roles in the recruitment and functionality of B cells under the inflammatory condition induced by ELL induction. Among them, decreased expression of *Cxcl13* and *C3* was found in the peritoneal macrophages by ELL, and their expression was enhanced by niclosamide ($p < 1.4e-29$; Fig. 8d). Moreover, B1a and B1b cells received the most *Cxcl13* signals, while B2 cells exclusively received the signals of *C3* (Fig. 8c). In addition, the expression of *Fn1* and *Ptprc* in macrophages was also altered by ELL and niclosamide ($p < 1.2e-5$), but the changes are limited (Fig. 8d).

The transcriptomic changes caused by ELL and niclosamide to B cells were also analysed. Compared to the macrophages, the overlaps of biological processes that were disrupted by ELL (ELL/sham) and reversed by niclosamide (ELL_N/ELL) were much fewer (Fig. 8e, f; Supplementary Data 6). Only 28 out of 114 biological processes that were enhanced by ELL were reversed by niclosamide (Fig. 8f; Supplementary Data 6). The limited overlaps include biological processes of "response to growth factor" and "homoeostasis of the number of cells" (Fig. 8g), which may be associated with the recruitment and maintenance of the number of B cells through macrophages.

These results suggest that ELL disrupted the communications between "large" types of macrophages and B1/B2 cells, which were rescued by niclosamide through up-regulating the expression of *Cxcl13* and *C3*. The transcriptomic alternations caused by niclosamide to B cells are minimal, indicating that macrophages are possibly the direct targets of niclosamide.

## Discussion
Niclosamide is an FDA-approved oral anthelmintic drug that is originally used to treat human tapeworm infections[29]. In addition to this common use, many clinical studies are ongoing to repurpose niclosamide for the treatment of other diseases such as different types of cancer, bacterial and viral infections, neuropathic pain, systemic sclerosis, and metabolic diseases[23,30–36]. Though the clear and direct binding targets of niclosamide have not been identified, studies have shown that niclosamide affects multiple important signalling pathways. One of the most appreciated action mechanisms is that niclosamide acts as a protonophore and thus uncouples oxidative phosphorylation and affects pH balance in cells[23,33,34]. In addition, signalling pathways of mTOR, Wnt/β-catenin, STAT3, NF-κB, and Notch are also modulated by niclosamide[31,32,35,37–39]. In endometriosis, we also found that niclosamide reduced the growth of lesions and decreased inflammation at lesion sites through its suppression of STAT3 and NF-κB signalling[24,40,41]. Moreover, we also found that niclosamide does not disrupt reproductive functions in mice, making it a relatively safe drug for treatment, which may preserve fertility, while common current treatment options do not[24].

Myeloid cells, especially macrophages, have been recognized as the central components within the endometriosis microenvironment[42]. Abnormal activation and increased numbers of macrophages along with elevated levels of proinflammatory cytokines such as IL-1β, IL6, IL8, and TNFα are present in the peritoneal fluid of patients with endometriosis[11,43,44]. Moreover, analysis of macrophages within the pelvic cavity and endometriotic lesions of patients has

demonstrated enhanced biological activities related to angiogenesis, neurogenesis, antigen processing and presentation, and tissue adhesion[45,46]. Our previous studies in mice have demonstrated that niclosamide treatment reduced inflammation in the peritoneal fluid, lesions, pelvic organs (uterus and vagina), and dorsal root ganglion[5]. In this study, we further reported that the majority of the transcriptomic changes in peritoneal macrophages, which are induced by ELL including enhanced activities of activation, angiogenesis, neurogenesis, and cell adhesion, were reversed by niclosamide. Therefore, niclosamide has a high potential to reduce the disease-promoting characterics of perturbed macrophages in patients with endometriosis.

Previous studies have suggested the long-term existence of transitory macrophages in the peritoneal cavity under inflammation in addition to traditionally recognized SPMs and LPMs[20,25,26], but our knowledge of those immature subtypes and their contributions to disease development is limited. In this study, we further characterized these subpopulations and identified three novel intermediate immature subtypes named IM1-3. IM1 cells were shown to be monocyte-derived and uniquely express *Folr2*, which encodes protein FRβ. FRβ+ macrophages have been reported to be activated and accumulated at the inflammatory sites of multiple diseases[47–49]. Deletion of FRβ+ macrophages has shown promising results for the resolution of several chronic inflammatory diseases such as rheumatoid arthritis and cancer[48–52]. However, some recent cancer studies identified FRβ+ macrophages as a subset of tissue-resident tumour-associated macrophages (TAMs), and these FRβ+ macrophages prevent the progression of cancer[53–57]. In the present study, IM1 cells also express high levels of *Trem2*, which is considered a marker for a monocyte-derived subset of TAMs. In addition, a subset of FRβ+ macrophages was also reported to be of monocyte origin[56]. Therefore, the origin of FRβ+ macrophages may vary in different tissues, and their functions in different inflammatory contexts may change, which awaits further investigation. The specific contribution of FRβ+ macrophages in the pathogenesis of endometriosis is also of great interest to explore further.

Severe inflammation results in the ablation of TIM4+ embryo-derived resident LPMs and increases the replenishment of these embryo-derived LPMs by TIM4- monocyte-derived macrophages[26]. Consistently, our study showed that niclosamide prohibited the replenishment of embryo-derived resident macrophages by both suppressing the maturation of monocyte-derived LPMs (IM2 and IM3) and preserving the population of embryo-derived LPMs. As the monocyte-derived LPMs were reported to have different functions from original TIM4+ LPMs[26], enhanced replenishment of these embryo-derived resident macrophages under the chronic condition may lead to long-term disruptions to the peritoneal niche of endometriosis.

The heterogeneity of macrophages in the human endometriotic microenvironment shares some similar characteristics with those found within the murine peritoneal cavity in this study. In the pelvic cavity and eutopic and ectopic endometrium of endometriosis patients, both CCR2+ monocyte-derived and tissue-resident macrophages were distinguished using scRNA-seq and traditional methods such as flow cytometry[45,46,58–60]. *FOLR2*+ macrophages were also identified in the endometriotic lesions in patients[46]. However, they have been considered as a resident subtype based on the expression of *FOLR2* and localization within cancerous tumours[56,61], which may not consider the variation of *FOLR2* expression in tissues of other locations. In addition, CD206+ (*MRC1*) dendritic-like cells and macrophages (correlated to Pre-SPMs, SPMs, and IM1) were identified in the pelvic cavity of patients[62]. The enhanced expression of CD206 (*Mrc1*)

and proinflammatory signature of macrophages in patient[62] is consistent with the enhanced expression of *Mrc1* and increased recruitment of CD206[+] (*Mrc1*) macrophages after lesion induction in our mouse model (Figs. 3c and 4e). In addition, by re-analyzing publicly available datasets of peritoneal exudate cells isolated from patients with or without endometriosis[45], we found consistent changes of some important genes in the peritoneal macrophages of patients with our data (Supplementary Fig. 7), including enhanced expression of *CCR2*, *SOCS6*, *KCTD12*, and *PTPRC*, and decreased expression of *TIMD4*, *APOC1*, *C1QB*, *PRDX5*, and *GNGT2*. Therefore, these alterations in patients with endometriosis are consistent with our findings using the mouse model, and most of those can be finely reversed by niclosamide treatment (Figs. 2; 3a–d and 4d, e).

CD5[+] B1 cells are widely present within the pelvic cavity of humans (correlated to B1a cells in mice) along with the presence of another newly identified subset of FceRIa[+] B cells[62–64]. Moreover, increased numbers of CD5[+] B1 cells and the FceRIa[+] subset were found within the pelvic cavity in patients with endometriosis along with the excessive production of auto-antibodies of B1 cells[63,65–69]. In the peritoneal cavity of mice, the replenishment of TIM4[+] embryo-derived macrophages with recruited cells decreased the recruitment of B1 cells due to lack of or reduced CXCL13 expression in the monocyte-derived macrophages[26,70]. Our studies further revealed that niclosamide rescued the expression of gene encoding intercellular signals secreted by macrophages including *Cxcl13, C3, Fn1* and *Ptprc*, and likely contributed to the regulation of B cells. However, the overall ratio of B cells within the pelvic cavity of humans is generally much lower than that of mice[62].

The subtypes of T cells within the pelvic cavity of humans are quite similar to those identified in the mouse model[62,71]. Both CD4[+] and CD8[+] T cells were found within the pelvic cavity in patients with endometriosis along with naïve and more differentiated subtypes[62,71]. Increased activation of T cells and the ratio of CD8/CD4 were also found within the pelvic cavity in patients[60,62,72,73]. However, our understanding of the function of B cells and T cells in the development of endometriosis and their associations with clinical symptoms and the severity of this disease remains limited. Due to the close intercellular communications between macrophages and B and T cells[27,74], dysregulated macrophages by lesion induction might be the leading cause of disrupted homoeostasis of the local immune microenvironment, and niclosamide could be a mediator to rescue their communication.

However, our understanding of macrophage subpopulations in the pelvic cavity and endometrium tissues and their subtype-specific contributions to the development of this disease in humans are still limited. A major caveat of this study is lacking validations of similar pathological processes of macrophage differentiation and replenishment identified in this mouse model also happening within human patients, which is critical for the effective targeting by niclosamide. In addition, the biological functions of important genes such as *Retnla*, *Cfb* and *Timd4* should be further explored by in vivo knockout studies to better elucidate the molecular targets and action mechanisms of niclosamide. Also, more studies are needed to understand the contributions of T cells and B cells, and their intercellular communications with macrophages for the pathogenesis of endometriosis in humans.

In summary, the heterogeneity and developmental characteristics of peritoneal macrophages were extensively explored this study using a mouse model of endometriosis. ELL enhanced the process of early differentiation and maturation of monocyte-derived macrophages while reducing the maintenance of embryo-derived resident LPMs. The increased replenishment of resident macrophages by monocyte-derived LPMs further disrupts the homoeostasis in the peritoneal cavity and affects the recruitment and functional activities of B cells. Niclosamide stepwise reverses the dynamic progression of recruited macrophages and preserves the population of embryo-derived LPMs, hence tuning the perturbed peritoneal microenvironment in endometriosis back to normal. Therefore, macrophages could be a direct target of niclosamide. Further studies examining the effective targeting of niclosamide on the pathogenesis of macrophages in human patients will help validate the proposed use of this drug as a new therapy for the treatment of this disease.

## Methods

**Animals and a mouse model of endometriosis.** All procedures were performed in accordance with the guidelines approved by the Institutional Animal Care and Use Committee of Washington State University (Protocol # 6751). C57BL/6 J mice were purchased from the Jackson Laboratory. Endometriosis-like lesions (ELL) were induced by inoculating syngeneic menstrual-like endometrial fragments from donor mice into the peritoneal cavity of recipient mice, as described previously[5,75]. Briefly, ovariectomized donor mice were primed with estradiol-17β (E2) and progesterone and induced decidualization by injecting sesame oil into uterine horns to produce a "menses-like" event. Then, decidualized endometrial tissues were scraped from the myometrium, minced, and injected i.p. into ovariectomized and E2-primed recipient mice under anaesthesia (50 mg tissue in 0.2 mL PBS per recipient). Sham mice were ovariectomized and E2-primed and injected with 0.2 mL PBS into the peritoneal cavity. Three weeks after the induction of ELL or sham, mice in the groups of sham, ELL, and ELL_N were orally administered with vehicle or niclosamide (200 mg/kg/day) for a total of 3 weeks (Fig. 1a), as described previously[5]. At 6 weeks following ELL induction, mice were euthanized, and peritoneal exudate cells were collected for further analysis.

**Preparation of peritoneal immune cells for single-cell RNA sequencing.** Peritoneal immune cells from three mice of each group were isolated and collected from the peritoneal fluid following our established method[5]. After removing red blood cells by lysis, remaining cell mixtures were used for cDNA library constructions following the manufacturer's protocol (10X Genomics, Inc.) of the Chromium Single Cell 3' Library & Gel Bead Kit V3[76]. All samples were multiplexed together and sequenced across one single lane of an Illumina NovaSeq 6000 S4.

**Single-cell data processing and analysis.** Raw data in FASTQ format were pre-processed with Cell Ranger V3.1.0 (10x Genomics) mapping to the mouse GRCm38/mm10 transcriptome to generate gene-cell matrices. A total of 13,859 cells from all three libraries were integrated into R using the Seurat package (V4.0.4)[77]. To filter out doublets and low-quality cells, criteria of 750,000 unique molecular identifiers (UMIs) and 500 genes per cell were set. In addition, cells with over 20% expression of mitochondrial genes were excluded for downstream analysis. Modified multivariate Pearson's RV correlations for each set of treatment replicates were calculated using the package of MatrixCorrelation (v0.9.2)[78]. The following correlations showed consistent sampling between libraries of treatments: ELL and sham = 0.962, ELL_N and sham = 0.979, ELL and ELL_N = 0.985, suggesting negligible batch effects between libraries. The "sctransform" function was then applied to normalize the remaining dataset with regression of mitochondria mapping percentage[79]. Dimensionality reduction was performed on identified variable genes by principal component analyses (PCA). The top 66 dimensions were selected for clustering ("resolution" set to 0.5) and uniform manifold approximation and projection (UMAP) visualization. Specific gene markers for each cluster were identified using the "FindAllMarkers" function. Differential gene expression between treatments was analysed using the "wilcox" test in the "Find-Markers" function with Bonferroni adjusted $p$-value < 0.05 showing significant differences. Gene Ontology (GO) and Gene Set Enrichment Analysis (GSEA) was performed with the R package, clusterProfiler V3.18.0, using all detected genes from the entire scRNA-seq library as background[80]. Terms were enriched with the nominal $p$-value < 0.05 and false discovery rate (FDR) ($q$-value) < 0.05.

**An interactive web tool to share scRNA-seq data of peritoneal immune cells.** Single-cell transcriptomic analysis has provided an unprecedented high resolution of peritoneal immune cells including different subtypes of B cells, macrophages, and T cells. To share our data with other researchers, we have created a cloud-based web tool for easy gene searches, which does not require complicated computer programming skills[81]. The webpage for this tool is: https://kanakohayashilab.org/hayashi/en/mouse/peritoneal.immune.cells/.

**Single-cell trajectory analysis.** The biological processes of "small" macrophage recruitment and "large" macrophage maturation were revealed by the trajectory analysis. Cells from clusters of "Pre-SPMs", "SPMs", and "IM1" or "IM2","IM3"

and "Timd4+ LPMs" were computationally selected in Seurat, and the two data matrices were imported, processed, and pseudo-ordered using the package of Monocle 3 in R following the standard pipeline[82].

**In silico knockout analysis**. Functional analysis of *Retnla*, *Cfb*, and *Timd4* was conducted using the R package of scTenifoldKnk V1.0.1[83]. A single-cell gene regulatory network (scGRN) was conducted using our scRNA-seq data from the sham group. Then, the expression of *Retnla*, *Cfb*, and *Timd4* was set to zero from the constructed scGRN to build their own corresponding "pseudo-knockout" scGRN. Perturbed genes by this virtual knockout were quantified by comparison of the "pseudo-knockout" scGRN to the original scGRN. Those significantly affected genes were used for GO analysis to show changes in biological processes caused by in silico knockout.

**Intercellular communication analysis**. Gene expression data of Seurat objects were used as input to model the probability of intercellular interactions between B cells and macrophages using the R package of CellChat V1.0.0[84]. The known database of interactions (CellChat.DB.mouse) between ligands, receptors, and cofactors was used as the reference.

**Re-analysis of one publicly available single-cell dataset of peritoneal macrophages**. A single-cell transcriptomic dataset of peritoneal macrophages from 19-weeks old female mice is publicly available[25]. These cells were selected based on the expression of CD11b with the removal of granulocytes and B1 B cells using flow cytometry. The raw data were downloaded from NCBI GEO (GSM4151331), pre-processed with Cell Ranger V3.1.0, and re-analysed in the R package, Seurat V4.0.4. For quality control, cells with the expression of fewer than 300 genes or over 5000 genes, and over 5% mitochondrial genes were excluded, resulting in a total of 4287 out of 4702 cells for downstream analysis. Following the standard pipeline of Seurat, the raw counts were normalized using a global-scaling normalization method ("LogNormalize"). Using the "ScaleData" function, the normalized dataset was further scaled for dimensional reduction. Based on the principal component analyses (PCA), the top 34 dimensions were selected for clustering and UMAP graphing. Trajectory analysis was performed on the cells computationally selected from the "Pre-SPM", "SPM", "IM1" or "LPM1", "LPM2", "LPM3" and "LPM4" as described above using the R package of Monocle 3.

**Flow cytometry**. Peritoneal exudate cells from three mice were pooled as one sample and used for analysing immune cell profiles by flow cytometry. A total of 15 mice were used for each group ($n = 5$). Briefly, the peritoneal lavages were centrifuged to collect peritoneal exudate cells. After lysing red blood cells by 1x RBC Lysis Buffer (BioLegend), an equal number of cells from each group were incubated at room temperature for 20 min with Zombie Aqua™ Fixable Viability dye (Bio-Legend) and blocked on ice for 20 min with Fc Block anti-CD16/CD32 (Thermo Fisher). Then cells were stained with fluorochrome-conjugated monoclonal antibodies (Supplementary Data 1) for 1 h. Samples were acquired with the Attune NxT Acoustic Focusing Cytometer using Attune NxT software (Invitrogen), and data were analysed with FlowJo v10.4. For analysis, only singlets (determined by forward scatter height vs. area) and live cells (Zombie Aqua negative) were used. Gating strategy for flow cytometry is shown in Supplementary Fig. 8.

**Quantitative Real-time PCR Analyses (qPCR)**. Total RNA of peritoneal exudate cells was extracted from distinct mice than those used for scRNA-seq for each group ($n = 6$) using TRIzol reagent (Sigma #T9424). cDNA templates were synthesized from 1 μg of purified RNA using the High-Capacity cDNA Reverse Transcription Kit (Thermo Fisher)[5,85]. qPCR was performed using a CFX RT-PCR detection system (Bio-Rad), and relative gene expression was evaluated by SYBR Green (Bio-Rad #1725274) incorporation. *Rpl19* was used as the reference gene to normalize mRNA expression levels. Data were analysed using the $2^{-\Delta\Delta Ct}$ method. Primer sequences were provided in Supplementary Data 2.

**Statistics and reproducibility**. For single-cell transcriptomic sequencing, a total of 3 mice from each treatment group were used for sample preparation. Pre-processing of raw sequencing data including transformation, normalization, and quality control were described above. For the analysis of differential gene expression, the default "wilcox" test was performed using the R package Seurat (v4.0.4). For flow cytometry, cells from three mice were pooled as one sample and a total of 15 mice were used for each group of treatments, thus ($n = 5$). For RT-qPCR of peritoneal immune cells, a new cadre of six mice (i.e., not the same mice used for scRNA-seq) for each treatment were used for RNA extraction ($n = 6$). For comparisons between three groups of treatments, one-way ANOVA followed by Tukey's multiple comparisons was used. Data were analysed with GraphPad Prism (version 9) and presented as means ± SEM. Statistical differences were indicated as *$p < 0.05$, **$p < 0.01$, ***$p < 0.001$, and ****$p < 0.0001$.

**Reporting summary**. Further information on research design is available in the Nature Portfolio Reporting Summary linked to this article.

## Data availability

The scRNA-seq data are openly available in the GEO database at NCBI, reference number GSE147024. All other data are available from the corresponding author upon request.

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

## Acknowledgements
Figure 1a was created using BioRender.com. The study was supported by the National Institutes of Health (NIH), Eunice Kennedy Shriver National Institute of Child Health & Human Development (NICHD) R01HD104619 (to K.H.).

## Author contributions
M.S. and K.H. designed experiments. L.Z. and M.S. performed experiments and analysed data. S.W. assisted with the webtool constructions and technical support with data analysis. J.A.M. and K.H. assisted in experiments and provided critical feedback on the manuscript. L.Z. wrote the paper. All authors read, edited, and approved the manuscript.

## Competing interests
The authors declare no competing interests.
