## [Peer Review File · Communications Biology]

Reviewers' comments:

Reviewer #1 (Remarks to the Author):

The authors have now addressed all of my concerns.

Reviewer #2 (Remarks to the Author):

The study elucidated the specific mechanisms by which niclosamide ameliorates endometriosis-like lesions in mice. The authors interestingly utilized currently available tools of in silico knockout and intercellular communication analyses. In addition, the authors sincerely addressed concerns that had been made by a preceding reviewer.

The amount of data is sufficient and the findings on the regulatory mechanisms of macrophages modulated by ELL and niclosamide are really intriguing, but the study is highly confined just to describing the transcriptomic changes or the trajectories of macrophages 'in the present model'. In silico knockout tool and CellChat package should be used to design further experiments to verify the obtained hypothesis. Therefore, it will improve this study for the authors to conduct additional experiments which suggest any significant functions of the focused genes (e.g., *Retn1a*, *Cfb*, and *Timd4*) in vitro or in situ with the present model, or with any human samples. Although the present mouse model has already been widely acknowledged, it should be utilized to validate authentic human endometriosis. In addition, the novelty of the study is limited because the efficacy of niclosamide in the present mouse model has already been demonstrated in the prior studies by the authors. In these respects, it is difficult to conclude that the manuscript in the submitted state meets the requirement for publication in Communications Biology.

The authors appreciate the comments and suggestions from the reviewers. Below we describe the point-by-point responses to the reviewer's concerns and comments. The verbatim comments of each reviewer are provided in **bold** followed by our response. All changes to the manuscript are indicated in the text by highlighting **red**.

Reviewer #2 (Remarks to the Author)

1. In silico knockout tool and CellChat package should be used to design further experiments to verify the obtained hypothesis. Therefore, it will improve this study for the authors to conduct additional experiments which suggest any significant functions of the focused genes (e.g., *Retnla*, *Cfb*, and *Timd4*) in vitro or in situ with the present model, or with any human samples. Although the present mouse model has already been widely acknowledged, it should be utilized to validate authentic human endometriosis.

Response: We have conducted in silico knockout of *Retnla*, *Cfb*, and *Timd4* in the population of macrophages and evaluated the consequences of these knockouts by the Gene Ontology analysis of affected genes (Fig. 4c, 6c & 7d). These in silico knockout data suggest important functions of these differentially expressed genes in the dynamic progression of macrophages after lesion induction, which leads us to conduct additional in vivo knockout models to further verify their pathological functions. We understand additional in vitro and/or in vivo analyses will further support our results. We are currently working to target multiple genes and generate new mouse lines to further understand the impact of macrophages in endometriosis pathophysiology. However, this would constitute a stand-alone study.

In addition, we found consistent changes in some of these important genes in the peritoneal macrophages of patients with endometriosis by re-analyzing publicly available datasets of peritoneal exudate cells¹. As shown in Supplementary Fig. 8, the gene expression of *CCR2* was significantly up-regulated in peritoneal macrophages in patients, which is consistent with their increased expression and the enhanced recruitment of Pre-SPMs and SPMs after lesion induction in our mouse model. Similarly, the gene expression of *TIMD4* and *APOC1*, marker genes for embryo-derived macrophages, were down-regulated in patients, verifying another important finding of our study that the population of embryo-derived macrophages was decreased by lesion induction. Therefore, there are some important consistencies between the pathological changes of macrophages in our mouse model and endometriosis patients. Please note that: only part of the dataset was uploaded by the authors and the low reading depth (mean: 11,000 reads/cell) prevents us from further analysis. We are currently working to generate our own single-cell RNA-seq data from patients. However, this would constitute a stand-alone follow-up report citing our current submission.

Furthermore, we provided more information in the Discussion section (Line 462-476) to compare the similarities of macrophage subpopulations and their pathological changes in mouse models and human patients, which should authenticate the clinical relevance of this mouse model.

2. In addition, the novelty of the study is limited because the efficacy of niclosamide in the present mouse model has already been demonstrated in the prior studies by the authors.

Response: This is not actually a limitation, but rather the rationale for the present study. We reported the effectiveness of niclosamide for decreasing inflammatory factors, alleviating pain signatures, and reducing lesion growth in our prior studies²⁻⁵, but the targets and working mechanisms of niclosamide were not revealed. In this manuscript, our evidence support that peritoneal macrophages are direct targets of niclosamide. By tuning the dynamic progression of peritoneal macrophages, niclosamide reduces the population of recruited macrophages while protecting embryo-derived macrophages and thus maintaining immune homeostasis in the local environment. Therefore, dysregulated peritoneal macrophages in patients could be targeted and tuned by niclosamide, which is promising, applicable, and of great novelty for the treatment of endometriosis.

1. Zou, G., *et al.* Cell subtypes and immune dysfunction in peritoneal fluid of endometriosis revealed by single-cell RNA-sequencing. *Cell & bioscience* **11**, 1-17 (2021).
2. Shi, M., *et al.* Efficacy of niclosamide on the intra-abdominal inflammatory environment in endometriosis. *The FASEB Journal* **35**, e21584 (2021).
3. Sekulovski, N., Whorton, A.E., Shi, M., MacLean, J.A. & Hayashi, K. Endometriotic inflammatory microenvironment induced by macrophages can be targeted by niclosamide. *Biol Reprod* **100**, 398-408 (2019).
4. Prather, G.R., *et al.* Niclosamide as a potential nonsteroidal therapy for endometriosis that preserves reproductive function in an experimental mouse model. *Biol Reprod* **95**, 74, 71-11 (2016).
5. Sekulovski, N., *et al.* Niclosamide suppresses macrophage-induced inflammation in endometriosis. *Biol Reprod* **102**, 1011-1019 (2020).

REVIEWERS' COMMENTS:

Reviewer #2 (Remarks to the Author):

I appreciate the effort the authors have made to address my specific concerns. I wholly checked the revised manuscript and the supplementary data added, and I totally understand the authors' points. However, my biggest concern is that although publicly available data and virtual knockout studies can facilitate analyses, the authors should consider real analyses for a couple of core statements. I personally think that at least the role of Retnla, Cfb, and Timd4 in macrophages should be validated in vitro.

In addition, the study lacks direct validation to demonstrate the existence of three novel subtypes IM1-3 in humans. Endometriosis is basically a human (or sometimes non-human primate) specific disease, and considering the lack of validation by human samples, the title should at least include "(e.g.) in a mouse model of endometriosis".

The authors appreciate the comments and suggestions from the reviewers. Below we describe the point-by-point responses to the reviewer's concerns and comments. The verbatim comments of each reviewer are provided in **bold** followed by our response. All changes to the manuscript are indicated in the text by highlighting **red**.

Reviewer #2 (Remarks to the Author)

1. I appreciate the effort the authors have made to address my specific concerns. I wholly checked the revised manuscript and the supplementary data added, and I totally understand the authors' points. However, my biggest concern is that although publicly available data and virtual knockout studies can facilitate analyses, the authors should consider real analyses for a couple of core statements. I personally think that at least the role of *Retnla*, *Cfb*, and *Timd4* in macrophages should be validated in vitro.

Response: We appreciate the suggestions from the reviewer. We have learned from single-cell level transcriptome analysis that the complex heterogeneity of macrophages and their functions change from dynamic progression and differentiation in the peritoneal environment. The study of in vitro knockout of *Retnla*, *Cfb*, and *Timd4* in macrophages was not provided in this manuscript because the phenotypes of primary macrophages will change in vitro environments, which may not be suitable for understanding the function of these genes for macrophage dynamics in vivo. We are currently working to generate new mouse lines to better understand their functions for the dynamic progression of macrophages and for the development of this disease. We are also collecting peritoneal exudate cells of human patients with/without endometriosis and will further evaluate the function of these important genes in the peritoneal macrophages of patients depending on the disease progression and symptoms. However, these additional experiments would constitute stand-alone studies.

2. In addition, the study lacks direct validation to demonstrate the existence of three novel subtypes IM1-3 in humans. Endometriosis is basically a human (or sometimes non-human primate) specific disease, and considering the lack of validation by human samples, the title should at least include "(e.g.) in a mouse model of endometriosis".

Response: Thank you for the suggestions from the reviewer. The title has been changed to "Niclosamide targets the dynamic progression of macrophages for the resolution of endometriosis in a mouse model". We have also toned-down our conclusion throughout the manuscript.